# Single-Wire Control and Fault Detection for Automotive Exterior Lighting Systems

**DOI:** 10.3390/s23146521

**Published:** 2023-07-19

**Authors:** George-Călin Seriţan, Costel-Ciprian Raicu, Bogdan-Adrian Enache

**Affiliations:** 1Department of Measurements, Electrical Apparatus and Static Converters, University “Politehnica” of Bucharest, 060042 Bucharest, Romania; george.seritan@upb.ro; 2Doctoral School of Electrical Engineering, University “Politehnica” of Bucharest, 060042 Bucharest, Romania

**Keywords:** lighting, automotive, LED, diagnostic, headlights, combination lamp

## Abstract

The design of exterior lighting is crucial for automotive manufacturers to ensure the visibility and safety of the driver. This article proposes a new strategy to control and diagnose one or more exterior lighting functions in electric vehicles by maximising the electrical faults that are detected and their transfer over a single-wire. The outcome is a decreased system cost and an additional method for vehicle lighting infrastructure control and diagnosis. Virtual simulation tools are used to explore the correlation between master-slave architecture and the effectiveness of the single-wire approach to comply with safety and regulatory demands. Safety-related and non-safety-related needs are explored to properly assess lighting functions, internal logic, and fault-case scenarios. Furthermore, assessing the viability of minimizing wire harness utilization while retaining the diagnostic capabilities for the controlled lighting sources, thereby simultaneously reducing the vehicle’s overall weight. This approach aims to concurrently decrease the overall weight of the vehicle. This work has three main contributions: (1) the development of efficient and reliable lighting systems in electric vehicles, a critical factor for achieving optimal performance, ensuring customer satisfaction, meeting regulatory compliance, and enhancing cost-effectiveness in automotive lighting systems. (2) Framework for future development and implementation of lighting systems in electric vehicles. (3) Simulation of the hardware architecture associated with the system strategy to achieve the desired system strategy for effectively applying the single-wire approach.

## 1. Introduction

The automotive industry’s scope for CO_2_ reduction and higher illumination quality, driven by increasingly stringent global regulations, has compelled significant investment in the design and development of exterior lighting. Light-emitting diode (LED) solutions have emerged as a key focus for meeting these objectives. LED lighting possesses attributes well-suited to long life, high reliability, excellent vibration and shock performance, optimised power consumption, and a compact package for the optical design. The rapid adoption of vehicle electrification, coupled with advancements in solid-state lighting sources, has established a solid foundation for the widespread implementation of LED-based exterior lighting. These solid-state light sources provide enhanced efficiency in terms of thermal management, optical performance, and overall reliability. The design and development of external lighting equipment types are continuously expanding, aiming to achieve greater flexibility across hardware, software, and optics to meet the demands of emerging functions like Adaptive Frontlight Systems (AFS) and comply with ECE regulations [1,2,3,4].

As the primary light source exterior vehicle lighting equipment utilizes either solid-state lighting sources or traditional bulbs. The former category features the latest vehicle architectures, while the latter appears in older or more affordable vehicles. Solid-state lighting sources are silicon-based illumination sources, such as LEDs, LASERs, and OLEDs. The overall efficiency of solid-state lighting, regardless of the driver type, is influenced by the need for voltage and electrical current regulation. This regulation process impacts the overall efficiency of the system, which can range from around 50% for linear regulators to over 80% for DC/DC switching regulators. The utilization of DC/DC switching regulators presents the potential for significant efficiency gains, particularly in optimizing power efficiency for hybrid vehicles [5]. 

LED lighting has managed to bridge the gap in optical performance with lens losses and comply with regulations as per the United Nations Economic Commission for Europe (UNECE-R 123) requirements. The AFS and Glare-Free High Beam (GFHB) [6] are continually optimised to adhere to operational and safety standards, providing the best user experience for the driver and other road users. In the case of rear vehicle lights, the automotive industry has adopted a combination of lamps and signal lights to create the desired aesthetic appeal. This approach involves consolidating multiple wire controls for various functions while introducing illumination effects [7]. The trend in solid-state lighting sources is to replace all halogen bulbs on all vehicle types as technology advances and becomes more cost-effective. This has led to the consolidation of many wire controls for different functions and the introduction of communication protocols to more robustly address functional safety needs.

Detecting faults and failures in LED-based lighting components presents one of the most challenging aspects of system efficiency. The primary issue is the number of control and feedback loops over independent wire control, resulting in a complex deadlock. In fig [8] an Intelligent diagnostic/prognostic framework for automotive electrical systems is proposed. While this platform is mostly developed for fault diagnosis and failure prognosis of vehicle electrical power generation and storage systems, it also proposes a failure model and criticality analysis for electric loads such as LED lamps based on recorded currents and a particle filter framework. This approach needs dedicated current sensors and complex processing systems for fault detection and prediction, but the outcome is a reliable time-to-fault detection methodology. In [9], a different data processing technique is used to address the challenges of collecting representative training data for developing a deep learning based FDD model for automotive systems. The proposed methodology combines the advantages of LSTM (Long Short-Term Memory) and 1D-CNN (Convolutional Neural Network) classifiers to overcome their limitations. This methodology also requires a complex measuring system and historical data, but it increases the accuracy of fault prediction. A different approach is presented in [10], and it argues in favour of the use of model-based, data-driven techniques and a combined approach for fault diagnosis. It explores the challenges of sparse data availability for rare events like faults and demonstrates the benefits of combining different approaches. The study demonstrates the benefits of a combined approach that integrates model-based diagnosis and data-driven anomaly classifiers for fault isolation. It has the main advantage that it can be extended very easily to different vehicle subsystems, but each of them requires a unique model. Other approaches include Statistical Process Control (SPC), Predictive Maintenance Models (PMM), Time-Series Analysis (TSA), etc. SPC [11] uses statistical methods to monitor and control a process, identifying any significant variations that might indicate a fault. In the context of automotive systems, SPC could be used to track data over time from a specific sensor or system, with alerts triggered if the data falls outside of an expected range. PMM [12] are models that use historical data to predict future faults. PMM uses data from a fleet of vehicles to predict when a specific component is likely to fail, based on factors such as age, usage, and maintenance history. TSA [13] can be used to detect faults characterized by trends or cycles in the data over time and predict future values, potentially forecasting a fault before it occurs.

While it addresses a different automotive system, Ref. [14] examines fault detection, diagnostics, prognostics, and fault modelling in Heating, Ventilation, and Air Conditioning (HVAC) systems. It covers topics such as fault detection (FDD) methods, IoT and cloud technology, fault modelling techniques, challenges, and the future prospects of FDD systems. The classification of FDD techniques is of great interest and can be easily extended to our study, making the proposed approach a data-driven technique part of a qualitative model-based method. The main advantage of these methods is their low-cost implementation and the lack of need for historical data, which comes at the cost of relative low accuracy. 

This paper proposes a new strategy to control and diagnose one or more exterior lighting functions in electric vehicles with a single-wire instead of many. As the backbone of the electrical and electronic architecture, the power supply, control, and diagnostics are critical components in modern automotive lighting systems. Due to their improved quality, reliability, and efficiency, light-emitting diodes (LEDs) have become the default choice for automotive lighting systems. This study employs virtual simulation tools to investigate the relationship between master-slave architecture and how the single-wire approach satisfies safety and regulatory demands. Through the analysis of internal logic and fault case scenarios, the article presents safety-related and non-safety-related lighting functional approaches, demonstrating the benefits of the single line approach. This work has three main contributions:To develop efficient and reliable lighting systems in electric vehicles, which is crucial for optimal performance, maintaining customer satisfaction, ensuring regulatory compliance, and ensuring cost-effectiveness in automotive lighting systems.To guide future development and implementation of lighting systems in electric vehicles.To propose a hardware architecture that can employ the one-wire fault detection method and validate it through simulation.

This paper is organised into four sections. Section 1 focuses on current market trends and directions, followed by an overview of automotive lighting system topologies in Section 2. Section 3 examines the electrical and electronic architecture of the subsystem and its bottlenecks. Section 4 presents a holistic approach and model-based on Simulink, demonstrating the diagnostic approach as an alternative to communication protocol solutions or current sensing, focusing on topology adjustments and gains. The paper concludes with a relevant and up-to-date list of references.

### 1.1. Front-Lighting

Conventional incandescent lamps used in automotive headlights do not require the utilisation of specialised electronics. Headlights based on LED, HID (high-intensity discharge), or Xenon require dedicated drivers to charge the loads and achieve start-up. The dedicated drivers boost the input voltage level up to the LED string voltage, more than 12 V, and up to thousands of volts to generate the electrical arc for HID or xenon. Regarding efficiency, LED loads exhibit minimal radiation losses but experience significant heat losses, as presented in Table 1.

Front-lighting system topologies vary depending on desired architectural constraints and the composition of features. Basic lighting can be driven directly with wire control, as shown in Figure 1a, while advanced features need the implementation of communication protocols to attain the desired system behaviour. Advanced front-lighting features on luxury vehicles are incorporated into the ADAS (advanced driving assistance system) to mitigate glare for participants in oncoming traffic, as seen in Figure 1b. Automatic front-light systems (AFS) enhance user visibility by managing beam projections. They occur when the vehicle engages in maneuvers and other participants are involved in traffic. The headlamp beams dynamically adjust to enhance the perception of road turns by other vehicles and reduce accidents involving pedestrians or other objects [1]. The front-lighting system contains functions such as:High beam (main beam);Low beam (dipped beam);Fog lamps;Turn indicators (direction indicator);Daytime running lights;Position lights;Side-markers;Auxiliary lamps, such as dedicated cornering lights or signature lighting.

Network-enabled headlights provide the capability of leveraging the use of high-resolution systems with symbol projection to achieve fully automatic control. Some of the functions are:Projection of navigation data (paths, lane changes, etc.);Dynamic lane assist (highlight of vehicle width on narrow passages);The highlight of warnings, road signs, and vehicle state.

### 1.2. Rear-Lighting

Rear lamps are the most crucial lights from a safety perspective and are designed to provide the state and dynamic behaviour intended to be conveyed by the vehicle user. Safety-related functions for rear lighting are direction indicators, brake lights, reversing lights, and fog lighting. Similar to front-lighting, solid-state lighting sources are used to leverage the regulatory constraints and add on the OEM (original equipment manufacturer)-specific light signature for branding enhancement, merged and controlled as well by ADAS features and passive safety needs such as brake light detection. They are linked with the complexity and overall vehicle needs, which need to be controlled either by wire, communication bus, or a mix to attain the required reaction time [15].

By design, they can be divided into six or more individual components to achieve the desired styling and surface coverage; one function, for example, the turn indicators, is used on the same side, with one section on the fender and another on the trunk. From a styling perspective, these components add a visually appealing look to the overall design. From a regulation perspective, dividing a function adds constraints for architecture control [16]. The arrangement may create the illusion that only a single lamp is installed on both the fender and trunk, giving the appearance of a single unit (Figure 2a), whereas on other designs, they look like distinct individual parts (Figure 2b).

### 1.3. Fault Detection Approach for Exterior Lights

For certain lighting functions, diagnostics or fault detection systems are mandatory to ensure compliance with regional regulations (for example, the Economic Commission for Europe, Federal Motor Vehicle Safety Standards, etc.) and safety compliance based on ISO2626262 [17]. Based on the technology and the control strategy, fault detection is performed either by the transmitter body unit or by the receiver body unit [17,18]; an example is shown in Figure 3, using the unified modelling language for low beam control.

Fault detection is utilized to promptly notify the driver in real-time in the event of any light failures, triggering the system to enter a fail-safe state. Light failures refer to LED malfunctions, power supply losses, and common electrical issues like open loads, short circuits, and internal failures within driving components. The electronic control unit internally conveys and assesses these failures through diagnostic tests. For instance, in the case of LEDs, the electronic control unit compares the actual current consumption and voltage across the LED string with pre-defined, coded values established during the design phase. This meticulous evaluation ensures the precise detection of any anomalies or deviations from expected performance. This ensures regulations set the required visibility targets, and address safety requirements, and take proper measures. For the most critical functions, such as low beams, if the control between the transmitter body unit—vehicle controller unit—and the lighting unit is lost, the safe state of the lighting unit is the low beam activation, and they remain active until control is reestablished, ensuring continuous functionality [17,18]. With CAN communication, end-to-end protection is used, as well as other information for plausibility checks such as vehicle speed. When wire control is used, the transceiver unit can diagnose the receiver body unit, which measures the output current. Hence, direct fault detection or the receiver body unit provides wired feedback to the system; if this feedback is lost, the vehicle considers the fail-state criteria as valid. In practice, achieving direct fault detection by the transceiver unit can be challenging due to losses in the harness or the input filter of the receiver body unit and losses in the transceiver output. Open load diagnostics are difficult to manage when the output current is higher than 50 mA; wired feedback is commonly preferred due to its reliability and ability to mitigate these challenges.

## 2. Fault Detection and Control Architecture for the Exterior Lighting Domain

When the same load is used for the same function, the control architecture for driving the lighting can be implemented either through buses or wired control systems. The first approach is usually the simplest to adopt, but it comes at a higher cost due to the need for a dedicated transceiver and software integration. In contrast, the wired solution is more optimised from a cost perspective and can be driven in an analogue manner [19]. A cross-functional flow of the lighting control architecture is presented in Figure 4 to illustrate the basic interaction between the user and the load.

The vehicle control unit receives user requests, which are acquired through steering column switches or by environment sensors such as rain, light, or objects in complex architectures. It compares that request with the previous states of the same driving cycle. The lighting controller unit receives the request and based on the load status, proceeds to restart and turn on the lights. If a fault is detected, it stores the data trouble code (DTC) and provides the status to a higher level, and the user is notified with either a telltale or a message in the dashboard for the light status, active or in failure. According to regulations, not all lights require a dedicated telltale indicator. Hence, some data are displayed on the dashboard with a generic message to check the light(s).

Electrical and electronic architecture control, for the same load utilisation manages two different functions, for example, daytime running lights (DRL) and front position lights (FPL) on the front side of the vehicle. Or brake light (or stop lamps) and rear position lamps (RPL) on the rear side [20], in some generic applications, is similar to Figure 5.

The electrical and electronic architecture must incorporate driver ICs utilising field-effect transistors (called smart FETs). These components detect typical output failures, including open loads and short circuits. The microcontroller drives the respective outputs based on pre-defined criteria or latches them for protection if needed. Based on Figure 5, on some architectures, function 1 is a switched power supply, and function 2 is pulse-width modulation (PWM) to inform the receiver body unit about the use case scenario to be used [21]. One of the limitations of this approach arises when the receiver body unit detects a failure. The lighting controller unit lacks the capability to sink or shut down the input, resulting in a consumption of less than 10 mA in such a way that the transmitter body unit and the vehicle controller unit cannot identify the open load criteria. In some cases, an additional power supply will be employed for the receiver body unit to address this issue. However, this solution needs the addition of extra wires to the diagram. The holistic control between the two units, while beneficial in some respects, may introduce unintended non-compliance with regulatory requirements, for example, by maintaining the DRL active while the Low Beams (dipped beams) or High Beams (main beams) are also active [20]. Another approach for solving the mentioned issue is to introduce wired feedback from the receiver body unit to the transmitter body unit, which is utilised to relay information about load states and functions, as reflected in Figure 6.

With the increased amount of wires, the probability of false status increases, and the complexity of defending them according to the system failure management and evaluation analysis (S-FMEA), resolving this issue is deemed highly challenging and often presents significant difficulties. When the function 1 status is failing, resembling a short circuit to the battery, while the actual load is working properly, the vehicle controller unit will assume the failure is true, and the user will be notified. For improved control strategy and cost reduction scope, some OEMs migrated towards a PWM feedback line to inform the transmitter body unit about the status of both functions in a more contained manner and immune to failure, like in Figure 7.

PWM-based codification provides a functional range for the function status, individually and with dedicated values: 0% duty cycle for the failure to the ground (SCC−), and 100% duty cycle for the short circuit to the battery failure (SCC+). The PWM coding can be customised to meet the specific requirements of the OEMs, particularly when multiple functions are combined on the same feedback line; the codification example is shown in Table 2.

An error margin or a locking feature should be integrated based on the PWM frequency usage, typically 100 Hz or 200 Hz.

The error margin in Table 2 is not defined and marked with x%, typically around 5%, to protect the line against EMI perturbances due to the proximity of other possible frequency active lines. The LED failures are more likely to be open load failures, around 80% of the time, than short circuit failures [22]; thermal stress or mounting impacts the reliability of these semiconductors.

## 3. Holistic Approach for Control and Failure Management Architecture on Exterior Lighting Domain

Issues and efficiency requirements as innovation drivers are continuously emerging in the automotive industry. Wire-harness layout and functional management are critical aspects that address these evolving needs. Exploring innovative system design and management approaches is crucial for achieving this. One approach is to integrate control and failure management into a single line, which results in a streamlined architecture with several benefits. Combining these two critical components in a smoother package improves system efficiency and performance. This innovation-driven need highlights the importance of continuous improvements and adopting new and innovative ideas to enhance system design and performance. Figure 8 provides an overview of this innovative approach and illustrates how it can contribute to achieving optimal efficiency in automotive lighting systems.

The proposed strategy was evaluated using Matlab-Simulink (version R2021a), using a constant current controller in the receiver body unit to regulate the stability and flux of the LEDs. An open load serial failure with the LEDs is generated to reflect the most common failure type. The overall schematic is shown in Figure 9. The proposed architectural approach can employ similar strategies to enhance efficiency across various scenarios.

In addition to the current controller, the receiver body unit contains a linear regulator control manager (LRCM), Figure A1 (Appendix A), to switch the load based on the desired function. In our case, one function is controlled with an 80% duty cycle and the second function with a 40% duty cycle at a frequency of 100 Hz for both of them. The LRCM replicates the PWM received and controls the LED load voltage. It comprises an output string voltage measurement to identify when an error occurs and replies on the PWM control line with the coded value. The waveforms presented in Figure 10, Figure 11 and Figure 12 highlight the model-based concept’s behaviour.

The evaluation of the proposed concept is aimed at demonstrating the compatibility and possibility of coexisting shared functionality over the same wire for input function control (FC) and output failure status (FS). In Figure 10c, gaps and additional margins for EMI behaviour are not considered. These gaps always represent an issue if the signals overlap. They must be handled in the final hardware concept, the limp-home mode concept, and the default duty cycle. These measures are needed in case of failure or if the function(s) must stay in a default state, such as being forced active or forced active at 30% of the maximum flux. The voltage spike during a failure (Figure 11c with green) is used to assess the open load state, as it is one of the most common failures encountered for the LED strings. Based on this, the PWM failure state (FS) is generated. The short-circuit failure rate of LEDs is relatively lower than open load failures. For a short circuit of the LED in a series topology, there is usually a need to have more than two failed LEDs to detect the load issue. For parallel topologies or grids, the regulation constraints of n-1 are used to achieve a lower probability of function loss. The LED radiant power is shown to have a constant value in Figure 11b for either duty cycle 80% or 40%; this is the instantaneous value and should not be misleading. The average radiant power versus the controlled duty cycle input is shown in Figure 12.

The transmitter body unit and receiver body unit input and output stages were designed to reflect the indent concept scope and functionality. In contrast, a comprehensive model approach would contain a push-pull topology for the control line to segregate the possible overlap in Figure A2 (Appendix A). A timer in each unit would be a mandatory strategy to sync the two units.

Cost-effectiveness is associated with the reduction of the harness and the number of pins on both the transceiver and receiver electronic control units. In Table 3, for one lamp and one vehicle and an assumption of a 5 m length and 5 mm diameter for copper wire at a rate of 0.15 €/kg, the paper reflects the impact of the proposal.

The solution adoption on a vehicle platform is based on an expected amount of vehicles sold of one million; hence, the cost reduction between the architecture with dedicated PWM (Figure 7) and the single-wire model (Figure 8) is 520 k€.

The proposed innovative work in this paper, with the scope of reducing costs and enhancing efficiency in automotive systems, has the potential to offer significant benefits for vehicle manufacturers and suppliers of the equipment. Although the cost savings will vary for each manufacturer due to their unique architecture and strategies, reducing pins, wire harnesses, and vehicle weight will substantially impact the industry. This state-of-the-art work can be particularly effective for OEMs that sell a large volume of vehicles. Although additional electronics may be required for adoption, the standardisation of the block control over time will help balance investment amortisation over time. The resulting reduction in the bill-of-materials, associated costs, and improvements in efficiency provide significant benefits for suppliers and manufacturers alike, making this an innovation-driven approach worth considering for future automotive systems.

## 4. Conclusions

The holistic model, the single-wire approach from this article, has the control and diagnostic capability shared and provides valuable insight for enhancing the overall safety and efficiency of the automotive systems. With no functional degradation, safety failure or regulation compliance failure, it portrays advantages for moving towards a more efficient approach. The lighting system design and control are studied and simulated to conclude the findings by contrasting the behaviour of control and diagnostic capability between the current models used and the hybrid proposal. The system approach for sharing the responsibility between the receiver and transmitter body units provides a sensing failure-free environment where one unit takes the overall responsibility when the other is in degraded mode.

Simulations carried out revealed the overall system structure and behaviour to attain the expected functionality desired for the intended scope of supervising the lighting sources and coding with the duty cycle in a push-pull communication, the sensed state. Based on the findings, we have concluded that no degradation is encountered by reducing the number of wires and sharing the responsibility at the system level.

In the future, we may consider extending the incipient fault detection technique presented by Deep Principal Component Analysis (DPCA) using the framework presented in [23].

## Figures and Tables

**Figure 1 sensors-23-06521-f001:**
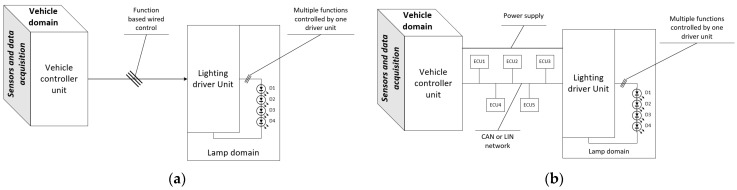
System topology control for exterior lighting with (**a**) wired control between transmitter and receiver units and (**b**) communication-based control between transmitter and receiver units.

**Figure 2 sensors-23-06521-f002:**
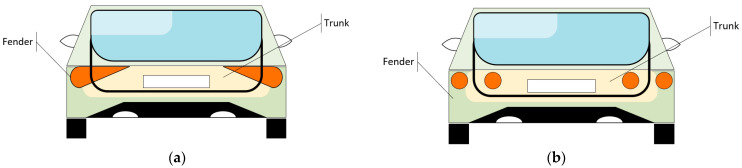
Configurations of rear lighting sources used to fulfil styling requirements and regulatory needs: (**a**) styling as one part on both the fender and trunk; (**b**) styling as individual parts on the fender and trunk.

**Figure 3 sensors-23-06521-f003:**
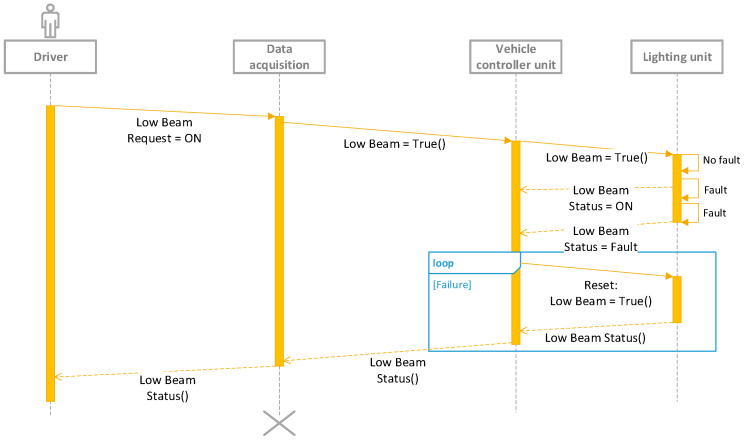
UML diagram of low beam control—simple use case.

**Figure 4 sensors-23-06521-f004:**
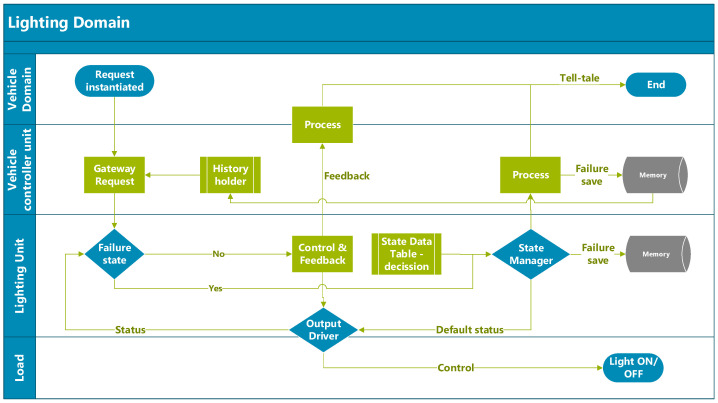
Cross-functional flowchart of lighting control—simple use case.

**Figure 5 sensors-23-06521-f005:**
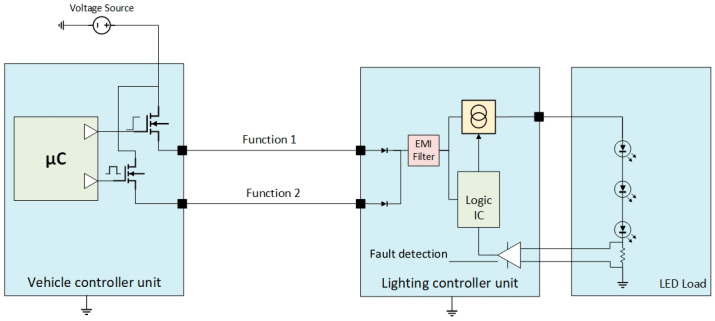
Electrical and electronic architecture design control for lighting’s combined function.

**Figure 6 sensors-23-06521-f006:**
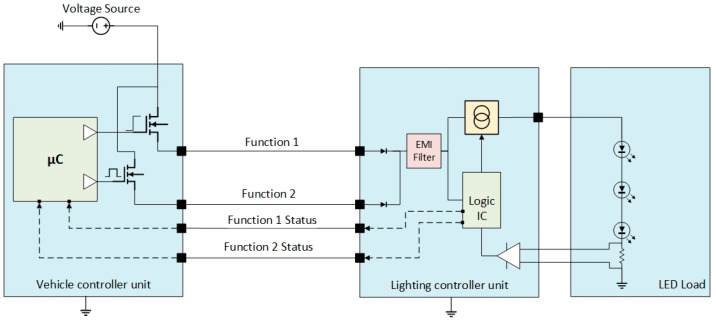
Electrical and electronic architecture for a combined function with dedicated wired feedback status.

**Figure 7 sensors-23-06521-f007:**
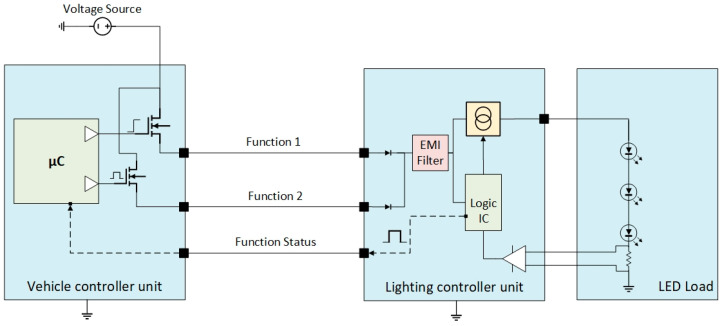
Electrical and electronic architecture incorporating a combined function and dedicated status feedback mechanism utilising PWM.

**Figure 8 sensors-23-06521-f008:**
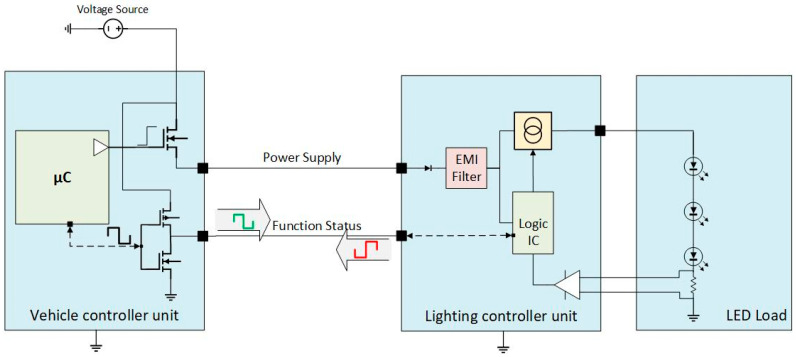
Electrical and electronic architecture for a combined function with combined PWM control and feedback return.

**Figure 9 sensors-23-06521-f009:**
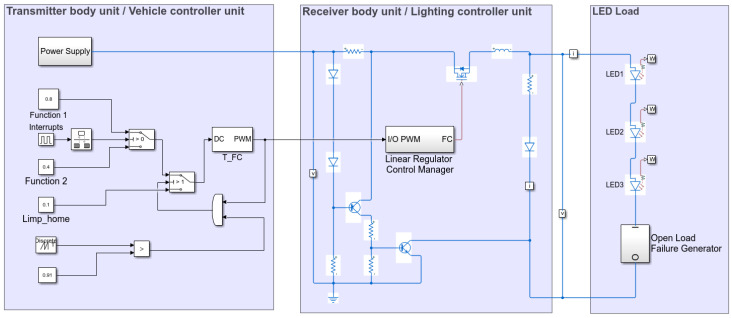
Simulink model for the electrical and electronic architecture of a combined function with PWM control and feedback over a single-wire.

**Figure 10 sensors-23-06521-f010:**
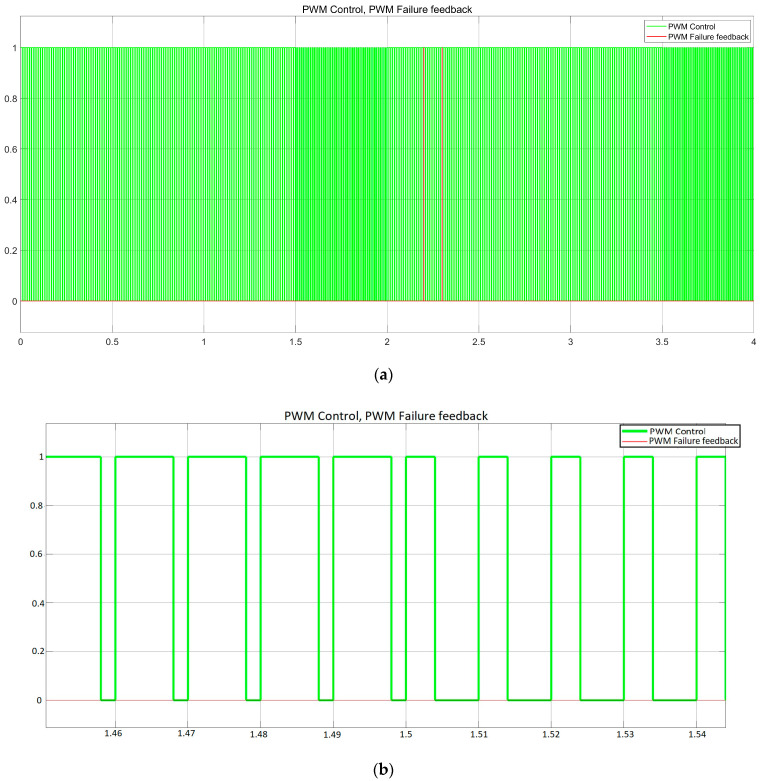
Representation of function status: (**a**) signal line data representation overview; (**b**) PWM control with a focus on duty cycle switch between 80% and 40%; (**c**) PWM control and feedback with a focus on the switch between control duty cycle at 80% (normal) and feedback duty cycle at 20% (failure).

**Figure 11 sensors-23-06521-f011:**
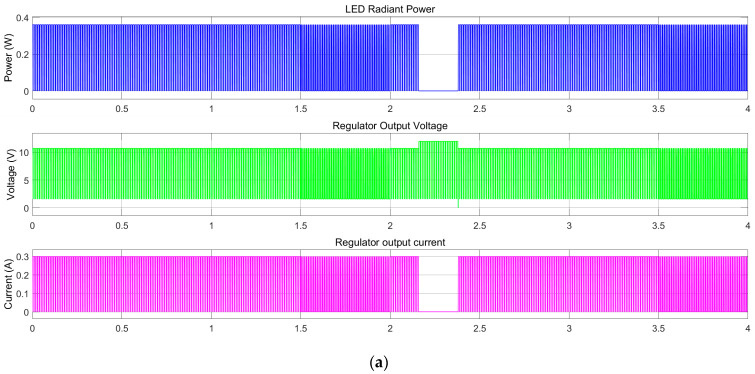
Current regulator and LED load output: (**a**) overview of regulator output and LED characteristics; (**b**) focus on the duty cycle switch between 80% and 40%; (**c**) focus on the switch between the control duty cycle at 80% and the failure reaction.

**Figure 12 sensors-23-06521-f012:**
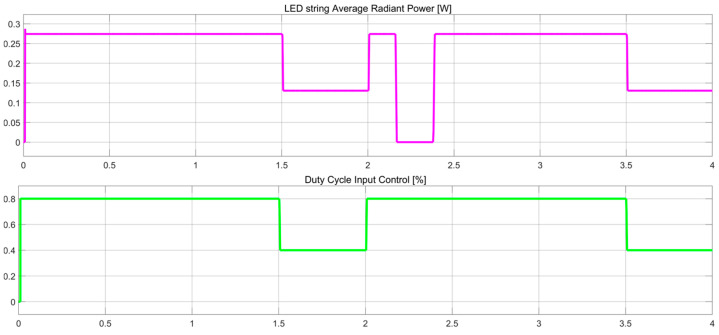
LED load average optical power output and PWM input control for receiver body unit.

**Table 1 sensors-23-06521-t001:** The efficiency of lighting loads [8].

Light Source	Loss in Radiation [%]	Heat Loss [%]	Luminous Efficacy [Lm/W]
Incandescent Lamp	81–86	5–6	6.7–20.7
HID (mercury)	62–65	16–22	100
HID (metal halide)	57–74	7–20	100
HID (sodium)	47–63	10–23	100
LED	0–0.2	80–88	20–130

**Table 2 sensors-23-06521-t002:** PWM codification usage—example case.

Light Function	Function 1	Function 2	Function Status
0%	ND ^1^	ND	SCC−
100%	ND	ND	SCC+
20% ± x%	FAIL ^2^	OK ^3^	20% ± x%
40% ± x%	OK	FAIL	40% ± x%
60% ± x%	FAIL	FAIL	60% ± x%
80% ± x%	OK	OK	80% ± x%

^1^ ND—not defined—the real state of the function cannot be determined; ^2^ FAIL: function is in failure state—Open Load (OL) or SCC−/+; ^3^ OK: function performs as expected.

**Table 3 sensors-23-06521-t003:** Cost impact for the solution when compared with the current market approach.

Architecture as in:	Number of Wires	Number of Pins	Weight [g]	Cost [€]
Figure 6	4	8	3.48	0.522
Figure 7	3	6	2.61	0.391
Figure 8 Single-wire model	2	4	1.74	0.261

## Data Availability

Not applicable.

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
