# Peer review of "Single-Wire Control and Fault Detection for Automotive Exterior Lighting Systems"

_sensors, 2023, doi:10.3390/s23146521_

Round 1
Reviewer 1 Report
The article titled,” Single-wire control and fault detection for automotive exterior lighting systems” was submitted to Sensors. This article proposes a new strategy to control and diagnose one or more exterior lighting functions, in electric vehicles, by maximizing sensing of different faults to take the appropriate actions over a single wire, with the outcome of decreasing the system cost.
I feel this article cannot be accepted for publication due to poor writing and technical flaws. However, I recommend some comments for improvement.
· Based on the numerical simulation, the proposed cost analysis is not feasible. What are the feasible implications for this study?
· The resolution of the figures is too low. It is difficult to read the legends. Please re-draw all the images with enlarged fonts.
· The introduction section is too narrow. With only a few references, it does not cover the scope of the proposed work.
· The conclusions section needs to be concise. I suggest adding a summary before the conclusion.
· Check the typos and grammatical errors. Such as superscripts, subscripts, etc.
I don't recommend acceptance based on the quality of the English language.
Author Response
We want to thank the reviewers for their valuable time and insightful comments on our paper. We truly appreciate the thoroughness with which you evaluated our work. Your constructive comments and suggestions have undoubtedly strengthened the quality and clarity of the paper.

Reviewer 2 Report
The authors investigated thee single-wire control and fault detection for automotive exterior lighting systems, and three main contributions are made. It is interesting and novel. Some minor revision can be addressed as below.
1. In this manuscript, some typing and spelling errors could be checked and corrected carefully, such as ‘fto ensurethe’ in abstract.
2. Some advanced fault detection methods could be reviewed in introduction such as Coupled neurons with multi-objective optimization benefit incipient fault identification of machinery and A comprehensive review: Fault detection, diagnostics, prognostics, and fault modeling in HVAC systems.
3. Some figures and their legends are not clear, and please update them.
Author Response

(The authors gave the same response as above.)

Reviewer 3 Report
This paper proposes a new strategy to control and diagnose one or more exterior lighting functions in electric vehicles with a single wire instead of many. Overall, the paper is well written and organized with a proper length. There are some points that are not very clear and should be addressed in the revised version:
1. Please further polish English by native English speaker, in particular, the Abstract and Introduction.
2. The introduction gives a brief overview of Light-emitting diode (LED) solutions for vehicles, however, it lacks an overview of fault diagnosis techniques for such cases. This makes the literature study incomplete and hence it is difficult to judge whether the methodology is novel or not.
3. Section 2.3. Fault detection approach on exterior lights should be enriched. What is the fault indicator? How to evaluate whether the system is faulty or not? All these basic problems should be explained clearly.
4.Incipient fault detection is an important research issue which is widely studied on Electrical and electronic architecture combined systems. The authors should supplement some results on this aspect in your future work in Conclusion section:
[1] Deep PCA-Based Incipient Fault Diagnosis and Diagnosability Analysis of High-Speed Railway Traction System via FNR Enhancement. Machines, 2023, 11(4): 475.
Moderate editing of English language required.
Author Response

(The authors gave the same response as above.)

Round 2
Reviewer 1 Report
It can be accepted. However, please improve the quality of Figures 9, 10, and 11. The text is not visible.
The quality of English has been improved.
Author Response
We want to thank the reviewers for their valuable time and insightful comments on our paper—all the requested figures were redone to increase their visibility.
Reviewer 3 Report
In a general way most of my comments were answered by the authors. My overall opinion about this paper is quite good. The manuscript is well written and acceptable for publishing
Minor editing of English language required
Author Response
We want to thank the reviewers for their valuable time and insightful comments on our paper - we performed another grammar and style check-up with the help of an English teacher.